# Autism Spectrum Disorder from the Womb to Adulthood: Suggestions for a Paradigm Shift

**DOI:** 10.3390/jpm11020070

**Published:** 2021-01-25

**Authors:** Cristina Panisi, Franca Rosa Guerini, Provvidenza Maria Abruzzo, Federico Balzola, Pier Mario Biava, Alessandra Bolotta, Marco Brunero, Ernesto Burgio, Alberto Chiara, Mario Clerici, Luigi Croce, Carla Ferreri, Niccolò Giovannini, Alessandro Ghezzo, Enzo Grossi, Roberto Keller, Andrea Manzotti, Marina Marini, Lucia Migliore, Lucio Moderato, Davide Moscone, Michele Mussap, Antonia Parmeggiani, Valentina Pasin, Monica Perotti, Cristina Piras, Marina Saresella, Andrea Stoccoro, Tiziana Toso, Rosa Anna Vacca, David Vagni, Salvatore Vendemmia, Laura Villa, Pierluigi Politi, Vassilios Fanos

**Affiliations:** 1Fondazione Istituto Sacra Famiglia ONLUS, Cesano Boscone, 20090 Milan, Italy; lmoderato@sacrafamiglia.org; 2Department of Brain and Behavioral Sciences, University of Pavia, 27100 Pavia, Italy; pierluigi.politi@unipv.it; 3IRCCS Fondazione Don Carlo Gnocchi, ONLUS, 20148 Milan, Italy; mario.clerici@unimi.it (M.C.); msaresella@dongnocchi.it (M.S.); 4DIMES, School of Medicine, University of Bologna, 40126 Bologna, Italy; provvidenza.abruzzo2@unibo.it (P.M.A.); alessandra.bolotta3@unibo.it (A.B.); a.ghezzo@fondazionedanelli.org (A.G.); 5Division of Gastroenterology, Azienda Ospedaliero-Universitaria Città della Salute e della Scienza di Torino, University of Turin, 10126 Turin, Italy; federico.balzola@usa.net; 6Scientific Institute of Research and Care Multimedica, 20138 Milan, Italy; piermario.biava@gmail.com; 7Department of Pediatric Surgery, Fondazione IRCCS Policlinico San Matteo, 27100 Pavia, Italy; marcobrunero@hotmail.it; 8ECERI—European Cancer and Environment Research Institute, Square de Meeus 38-40, 1000 Bruxelles, Belgium; eburg@libero.it; 9Dipartimento Materno Infantile ASST, 27100 Pavia, Italy; alberto_chiara@asst-pavia.it; 10Department of Pathophysiology and Transplantation, University of Milan, 20122 Milan, Italy; 11Centro Domino per l’Autismo, Universita’ Cattolica Brescia, 20139 Milan, Italy; luigi.croce@unicatt.it; 12National Research Council of Italy, Institute of Organic Synthesis and Photoreactivity (ISOF), 40129 Bologna, Italy; carla.ferreri@isof.cnr.it; 13Department of Obstetrics and Gynecology, Fondazione IRCCS Ca’ Granda Ospedale Maggiore Policlinico, 20122 Milan, Italy; niccologiovannini@hotmail.com; 14Autism Research Unit, Villa Santa Maria Foundation, 22038 Tavernerio, Italy; enzo.grossi51@gmail.com; 15Adult Autism Centre DSM ASL Città di Torino, 10138 Turin, Italy; rokel2003@libero.it; 16RAISE Lab, Foundation COME Collaboration, 65121 Pescara, Italy; manzotti.andrea68@gmail.com; 17Medical Genetics Laboratories, Department of Translational Research and of New Surgical and Medical Technologies, University of Pisa, 56126 Pisa, Italy; lucia.migliore@unipi.it (L.M.); andrea.stoccoro@unipi.it (A.S.); 18Associazione Spazio Asperger ONLUS, Centro Clinico CuoreMenteLab, 00141 Rome, Italy; davide.moscone@cuorementelab.it; 19Neonatal Intensive Care Unit, Department of Surgical Sciences, Puericulture Institute and Neonatal Section, Azienda Ospedaliera Universitaria, 09100 Cagliari, Italy; mumike153@gmail.com (M.M.); vafanos@tin.it (V.F.); 20Child Neurology and Psychiatry Unit, IRCCS ISNB, S. Orsola-Malpighi Hospital, Department of Medical and Surgical Sciences, University of Bologna, 40138 Bologna, Italy; antonia.parmeggiani@unibo.it; 21Milan Institute for health Care and Advanced Learning, 20124 Milano, Italy; dott.pasinvalentina@gmail.com; 22Università Telematica Pegaso, 80134 Napoli, Italy; monicaperotti@virgilio.it; 23Department of Biomedical Sciences, University of Cagliari, 09042 Cagliari, Italy; cristina.piras@unica.it; 24Unione Italiana Lotta alla Distrofia Muscolare UILDM, 35100 Padova, Italy; dr.tizianatoso@gmail.com; 25Institute of Biomembranes, Bioenergetics and Molecular Biotechnologies (IBIOM), National Research Council of Italy, 70126 Bari, Italy; r.vacca@ibiom.cnr.it; 26Institute for Biomedical Research and Innovation (IRIB), National Research Council of Italy, 98164 Messina, Italy; david.vagni@irib.cnr.it; 27Department of Pediatric, Moscati Hospital, 81031Aversa, Italy; dotvendemmia@libero.it; 28Scientific Institute, IRCCS Eugenio Medea, Via Don Luigi Monza 20, 23842 Bosisio Parini, Italy; laura.villa@lanostrafamiglia.it; 29Neonatal Intensive Care Unit, Azienda Ospedaliera Universitaria, 09042 Cagliari, Italy

**Keywords:** Autism Spectrum Disorder (ASD), pathogenesis, prevention, epigenetics, immune activation, gut dysbiosis, mitochondrial impairment, oxidative stress, metabolomics, machine learning

## Abstract

The wide spectrum of unique needs and strengths of Autism Spectrum Disorders (ASD) is a challenge for the worldwide healthcare system. With the plethora of information from research, a common thread is required to conceptualize an exhaustive pathogenetic paradigm. The epidemiological and clinical findings in ASD cannot be explained by the traditional linear genetic model, hence the need to move towards a more fluid conception, integrating genetics, environment, and epigenetics as a whole. The embryo-fetal period and the first two years of life (the so-called ‘First 1000 Days’) are the crucial time window for neurodevelopment. In particular, the interplay and the vicious loop between immune activation, gut dysbiosis, and mitochondrial impairment/oxidative stress significantly affects neurodevelopment during pregnancy and undermines the health of ASD people throughout life. Consequently, the most effective intervention in ASD is expected by primary prevention aimed at pregnancy and at early control of the main effector molecular pathways. We will reason here on a comprehensive and exhaustive pathogenetic paradigm in ASD, viewed not just as a theoretical issue, but as a tool to provide suggestions for effective preventive strategies and personalized, dynamic (from womb to adulthood), systemic, and interdisciplinary healthcare approach.

## 1. Introduction

Autism spectrum disorder (ASD) is currently diagnosed on the basis of the clinical assessment of behavioral features [1], and is characterized by a wide spectrum of presentation and frequent association with medical comorbidities [2]. People with ASD show an increasing endophenotypic complexity suggesting a systemic disorder with multilevel health needs. The manifold phenotype and the dramatic increase in prevalence in the last decades—currently 1:54 in the USA [3]—require the conceptualization of a pathogenic paradigm able to explain both the clinical and the epidemiological findings.

In research, there is a need to take stock of the situation in two main issues. First, effective strategies aimed at reversing the course of prevalence are urgently required. Secondly, evidence concerning the biological complexity of ASD requests a coherent translation into the healthcare model and the clinical practice.

Fetal neural programming, occurring during the ontogenesis, and early live neuroplasticity are crucial events in neurodevelopment and identify the time window of maximum brain opportunity in the embryo-fetal period and in the first two years of life (the so-called ‘First 1000 Days’) [4].

In this period, exogenous insults and changes in the maternal milieu are expected to have the maximum disturbing effect and lifelong consequences on health. The dynamic molecular machinery involved in the ontogenesis transforms early life inputs into long-term programmatic outcomes, influencing the enzymatic and immuno-neuroendocrine pathways, which define the basis of the homeostasis during intrauterine and postnatal life [5].

Several biological abnormalities are involved in the etiopathogenesis of ASD. Our study reviews the main topics addressed in biological research—genetics, epigenetics, environmental issues, immunogenetics, immunology, microbiology, and metabolic and electrophysiological impairments—with particular reference to their synergistic interactions and their links with the clinical phenotype. Therefore, a dynamic (from the womb to adulthood) perspective and the concept of multisystem disorder seems to be the most plausible framework for the study of neurodevelopmental disorders.

Machine learning will also be proposed as the most suitable method consistent with the biological complexity of ASD.

The healthcare model currently aimed at supporting autistic people is quite fragmented, and often fails to integrate psychoeducational and biological interventions. Drawing a ‘fil rouge’—from the genome to the effector biochemical pathways, and from them to the multifaceted clinical phenotype in ASD—seems to be the premise for the long looked-for breakthrough in clinical practice.

The conceptualization of the pathogenetic paradigm is proposed as the premise for necessary changes in clinical practice, highlighting a common thread, from primary prevention in pregnancy, to the healthcare model addressed, to the complex lifelong needs of people with ASD.

According to a dynamic and systemic perspective, neurodevelopmental disorders seem to be better depicted by a trajectory of possible and, at least partially, modifiable frailty rather than by a static picture resulting from a fixed and inevitable brain damage.

Consequently, this paradigm encourages the best efforts for effective primary prevention strategies addressed to the ‘First 1000 Days’, in order to turn this time window into the best chance for human health. At the same time, evidence concerning numerous biologic abnormalities in ASD provide suggestions for some already feasible adaptations of the healthcare model addressed to autistic people.

### 1.1. Clinical Features of People with ASD: Not Only a Behavioral Disorder

Autism is an early onset—and in most cases, lifelong—neurodevelopmental disorder, currently diagnosed through standardized behavioral testing between the ages of 2 and 4 years. The broad range of presentation accounts for the standard definition of Autism Spectrum Disorder (ASD) as an alteration in social communication and interaction across multiple contexts, in conjunction with repetitive behaviors, a restricted pattern of interests, and sensory abnormalities [1]. In the most common early onset pattern, atypical features in vocal–verbal language and socio-communicative development are detected in the first 12 months, and failure in acquiring skills is reported. However, some children initially show a period of apparently typical development, followed by a loss of previously established skills. The phenomenon is termed ‘regression’ and typically occurs between 15 and 30 months of age, with a mean of 21 months [6]. There is no consensus regarding the prevalence of regression within autism and other ASD diagnoses [6]. The phenomenon certainly needs to be better understood as well as defined by symptoms, including regressive events that may occur later in life. A wide range of different combinations of behavioral characteristics, beside numerous emotional and cognitive features, contribute to the wide variability of the clinical picture and to the varied impact on major life areas [7] and quality of life [8]. The presence and severity of intellectual impairment is the most relevant characteristic influencing the outcome [9]. The wide range of combinations within the spectrum poses the difficult challenge in responding to this variegated group of people, endowed with unique needs and unique strengths, in a personalized manner.

The striking increase in prevalence in the last decades [3] demands deep reflection and evaluation. Increased awareness of the disorder and evolving diagnostic criteria have undoubtedly contributed to this result. On the other hand, a broad scientific consensus converges on the concept that change in diagnostic criteria is not enough to explain such a significant rise in the occurrence of ASD [10], and environmental factors are supposed to fill the gap.

Currently, many areas are engaged in research on ASD, including genetics, epigenetics, immunogenetics, immunology, microbiology, and biochemistry—the last, in particular, with regard to mitochondrial impairment and oxidative stress. As the submerged part of an iceberg, the biological complexity underlying behavioral abnormalities accounts for a systemic disorder, not limited to the brain, but involving other organs and systems as well. This is consistent with the frequent occurrence of comorbidities [2]—in particular, immunologic [11] and GI abnormalities [12]. They are not just ancillary characteristics, but rather the result of a systemic disorder, impacting on the health of people with ASD and requiring to be properly faced.

### 1.2. Genetics, Epigenetics, and Environment: Who’s to Blame?

Quantitative genetic studies, including twin studies, have suggested a role of genetics in ASD [13,14,15].

The genetic architecture in ASD is complex: Hundreds of gene variants, identified using genome-wide association studies (GWAS) [16] and copy number variants (CNVs) [17,18], as well as de novo mutations in non-coding regions, affecting transcriptional and posttranscriptional regulation [19], have been found associated with ASD. None of the individual genes identified to date account for more than 1% of ASD cases. Though many candidate gene-disease associations were suggested, molecular investigations have yet to identify a consistent association of ASD with biological markers [20]. This also reflects the complexity of the ASD phenotype, which overlaps with several neurodevelopmental/psychiatric disorders [21,22] and involves numerous non-neurological comorbidities [23]. ASD susceptibility evaluation using family-based genetic studies is now accompanied by population-based epidemiologic studies, which pointed to the role of environmental factors [24] in explaining the rate of non-genetic variance [25,26].

Altogether, the genetic data suggest the need to depart from a linear genetic model. The striking increase in prevalence in ASD makes implausible the hypothesis of an increasing number of monogenic diseases, and for this reason an epigenetic model has been suggested. Epigenetics refers to changes in gene regulatory mechanisms that are independent from alteration of the underlying DNA coding sequences [27]. Epigenetics tunes gene expression based on changes in the cellular environment, in an adaptive and predictive sense, predisposing cellular molecular equipment aimed at homeostasis [28].

DNA methylation, histone tails modifications, and non-coding RNAs (i.e., microRNAs) are the most commonly studied epigenetic effectors [29]. They influence the establishment of gene transcription patterns through multiple mechanisms, regulating the accessibility of genomic loci to a large number of regulatory factors (i.e., transcription factors, enhancers, silencers) as well as the expression/stability of mRNAs. Changes in epigenetic signatures during the developmental stage finely tune the differentiation of precursor cells into their specific mature state [30]. Therefore, epigenetic markers display a relatively high level of plasticity during periods of cellular differentiation, including neurodevelopment [31]. Since the embryo-fetal period and the first two years of life represent the temporal window of maximum neuroplasticity, environmental exposure occurring during pregnancy is expected to lead to long-term modifications in epigenetic patterns and to have maximum impact on neurodevelopment [32].

Synaptic plasticity and chromatin binding are the most important biological functions emerged from the analysis of hundreds of genes associated with ASD [33,34]. The enrichment of chromatin binding genes associated with ASD suggests their potential role in the etiology of this disorder. Rett syndrome is a well-known example of a genetic neurodevelopmental condition that includes autistic behavior, and whose etiology is directly related to epigenetic regulation. Rett syndrome is caused by mutations in MECP2 gene, which encodes for the methylated DNA binding protein MeCP2 [35], causing either the activation or the inhibition of gene transcription, depending on the genomic context [36]. Interestingly, beside Rett syndrome, also Fragile X syndrome, Angelman syndrome, and Beckwith-Wiedemann syndrome, are all caused by epigenetic dysregulation, and each one shares a phenotypic overlap with ASD [37,38].

Many epigenetic markers are differentially expressed in ASD. A recent systematic review by Dall’Aglio et al. [39] analyzed studies on epigenetic modifications found in ASD and other neurodevelopmental disorders, focusing on global and gene-specific methylation, as well as on epigenome-wide DNA methylation (EWAS) in brain and blood tissues, and on histone modifications in the brain tissue. A number of shared biological pathways of relevance to neurodevelopment, were reported by independent EWAS, including synaptic and neuronal processes, immune response processes, brain development, and cellular differentiation [39]. In candidate genes studies, the identification of differential DNA methylation in the proximity of three genes—PRRT1, C11orf21/TSPAN32, and OR2L13—was confirmed in multiple independent analyses. The function of these genes is still little known, anyway, they are probably involved in neurological disorders. Analogously, histone modifications (methylation and acetylation) in association with ASD were consistently found in the gene coding for H3K27 in the cerebellum and cortex of autistic patients [39].

Moreover, several studies evidenced that short non-coding RNAs, such as microRNAs, are differentially expressed in brain tissue [40] as well as in the periphery (i.e., serum/plasma, saliva) [41,42] of ASD patients compared to typically developing controls.

Therefore, moving from a traditional paradigm focusing exclusively on a linear genetic model, epidemiological and clinical findings in ASD suggest the conceptualization of a model integrating genetics (hardware), environment (information), and epigenetics (software). Thus, the focus is shifted from the highly phylogenetically conserved human DNA sequence to the misleading way by which the stored instructions are read. The following paragraph explains in detail the epigenetic paradigm.

### 1.3. Widening the Gaze: From ASD to the Epidemiological Transition of NCDs

The plausibility of the epigenetic paradigm regarding ASD is supported by a wider phenomenon involving numerous non-communicable diseases and disorders (NCDs), which showed a striking increase in prevalence in the last decades, in parallel with neurodevelopmental disorders [43,44].

We are witnessing a profound epidemiological transformation, which concerns immuno-allergic, inflammatory, metabolic, chronic-degenerative, neurodevelopmental, neuropsychiatric, neurodegenerative, and neoplastic diseases [45]. The phenomenon as a whole suggests a common pathogenetic model. As the time frame is too short for genetic changes to have had an appreciable impact on the prevalence of the abovementioned diseases, it is more plausible that the increase reflects changes in gene programming (epigenetics) induced by a growing number of environmental stressors during critical time windows in development [46,47]. This interpretation is the basis of the theory of the epigenetic/embryo-fetal origin of diseases (DOHaD—Developmental Origin of Health and Diseases) [48]. The DOHaD theory suggests a systemic perspective, in order to explain the reasons of the profound transformation of human health and disease. The theory takes into account the impact of environmental stressors on reactive–adaptive and predictive epigenetic modifications (fetal programming) in cell and tissue differentiation, with long-term consequences on individual development and transgenerational impact. Imperfect correspondence between embryo-fetal programming and postnatal environment (that is, mismatch between the prenatal prediction and the actual postnatal environment) might also contribute to the onset of NCDs [45].

Beyond the definition, in the essence, epigenetics means a new systemic genome model which places the DNA sequence in the center of a dynamic, fluid, unitary, and interactive molecular network, involving the inner and the outside environment [28]. The genome is proposed as a fluid system, made up of the DNA sequence, the responsive histone structure, and information from the surrounding environment, in the broader sense of the term. The epigenome—as a software switching genes on and off [28]—tunes the matching between information coming from outside (environment) and information codified by millions of years in the DNA (that is the hardware). Through mechanisms modulating the programming, transcription, and translation of the message, the epigenome orchestrates the “natural genetic engineering” for the structural and functional changes of cells and tissues, contributing to the evolving phenotype, both in physiological and pathological situations [49].

During pregnancy, the placenta “translates” the external environment to the fetus through epigenetic molecular adaptations, in particular by modifying the methylation of imprinted genes, which act as key controllers for fetal development. A wide range of maternal inputs (i.e., over and under-nutrition, smoking, drug and alcohol intake, environmental toxicants, infections, and stress) can induce changes in placental physiology, ranging from alterations in placental morphology and weight, to the more subtle changes in placental gene expression, involving important signals directed towards the fetus [50]. Imprinted genes display a variety of cellular roles, such as cell cycle control, ion channels, protein synthesis/degradation, and nutrient transport. They are maximally expressed during the prenatal and/or postnatal period, and predominate in tissues governing resource allocation (brain, placenta, adipose tissue, and pancreatic beta cells). Consistently with these findings, a substantial number of imprinted genes are critical for placental function and normal fetal growth and development [51].

Epigenetic programming is highly sensitive to changes in the cellular environment. Indeed, epigenetic regulation is a widely utilized adaptive mechanism, allowing cells to maintain a favorable metabolic status under different conditions, including exposure to physiological substances (e.g., hormones, neurotransmitters, or growth-regulation factors), xenobiotics (e.g., pollutants, toxic chemicals), or even infectious agents (e.g., bacteria or viruses, fungi or parasites) [52]. Epigenetic regulation may facilitate adaptation to changes in the cellular environment through stable alterations of cellular phenotype, potentially resulting in progressive differentiation and maturation during fetal and possibly postnatal development [53].

During the ontogenesis, epigenetic software is programmed in an adaptive and predictive sense, driving cellular differentiation and setting up the metabolic, immunologic, and endocrine pathways for lifelong homeostasis. As far as neurodevelopment is concerned, the ‘First 1000 Days’ are the period in which neuroplasticity is most lively, for neuronal proliferation, differentiation, migration, synaptogenesis, and pruning. In other words, this is the most vulnerable time window for the wiring of individual connectome [54,55]. For the same reasons, the ‘First 1000 Days’ offer a unique opportunity to provide the best setting for the best lifelong trajectory in neurodevelopment [4].

### 1.4. Maternal Immune Response in Neurodevelopmental Disorders

Mounting evidence suggests the existence of a link between immune function and neurodevelopmental disorders [56,57]. The immune system plays a fundamental role in brain development, both in the physiological and in the pathological trajectories. Immune response impacts on neuronal migration, synaptogenesis, white matter organization, and remodeling (pruning), that is, on some of the crucial steps of neural network development [58,59,60,61]. Therefore, it is not surprising that an abnormal immune response might influence neurodevelopment [57].

Maternal infections and autoimmune diseases provided the first demonstration of the possible impact of immune response on brain development during pregnancy.

In the period following the Rubella outbreak of the 1960s, ASD was diagnosed in 5–10% of children born to mothers that were infected by the Rubella virus [62,63]. The prevalence of ASD and schizophrenia was found to be significantly higher in children exposed to infection than in those not exposed [64]. Afterwards, similar correlations were found with influenza, measles, mumps, varicella, and polio epidemics [65]. Several prospective studies confirmed the association between maternal viral infections and the onset of neuropsychiatric disorders in children, and over the years the list lengthened with the inclusion of bacterial infections (tonsillitis, sinusitis, pneumonia) and parasites (Toxoplasma gondii) [64,66]. A Swedish research found a 30% increase in ASD when the mothers were hospitalized for viral infections during pregnancy, and a significant association was found between viral infections and ASD in offspring [67].

The recent outbreak of the Zika virus raises concerns about the increased risk of ASD, given the high number of babies born with microcephaly, structural brain abnormalities, and neurological alterations in regions affected by the virus [68]. Similar concerns relate to the current SARS-CoV-2 pandemic, since maternal cytokine storm (mainly, IL-6 and IL-17) and intra-uterine inflammation might interfere with the fetal epigenetic machinery [69]. In fact, beside the vertical transmission, maternal infections may impact fetal neurodevelopment through the maternal immune activation (MIA). Several studies reported an increased risk of ASD in the offspring of mothers with active autoimmune diseases during pregnancy. Particularly, exposure to psoriasis, systemic lupus erythematosus, rheumatic arthritis, and autoimmune thyroid disease could significantly increase the risk of ASD [70,71,72]. In mothers with autoimmune diseases a pathogenetic effect is suggested for maternal autoantibodies and cytokines crossing placental barrier and the brain–blood barrier [73,74]. A meta-analysis showed that maternal autoimmune diseases were associated with a 34% increased risk of ASD in offspring, compared with the control groups [75]. Correlations between postnatally ASD-specific maternal autoantibodies and maternal metabolic conditions during gestation have been reported [76].

Neuroinflammation could damage fetal brain tissue and exert adverse influence on brain development [56]. Cytokines, activated T cells, autoantibodies, and microglia—the macrophages of the central nervous system—exert a pivotal role on antigen presence and cytokines production [56,77].

Maternal autoantibodies targeted against the Folate Receptor α (FRα) deserve to be mentioned in this section. Folate plays a key role in neural development during the embryo-fetal period and the early years of life [78]. Folate plays an essential role in cell-to-cell communication and in purine, methylation, and redox metabolic pathways. Since its concentration in the brain is several folds higher than in plasma, it relies on active import mechanisms, based on FRα, to cross both placental and brain barriers, but the transport is inhibited by anti-FRα autoantibodies (FRAAs) [79]. Animal models demonstrated a correlation between the exposition of pregnant dams to FRAAs and ASD-like features in the offspring [79]. This is consistent with the finding of more severe ASD symptoms in autistic children from mothers positive for FRAAs [78]. The relevance of this mechanism is confirmed further in life by the higher prevalence of FRAAs in children with ASD compared to controls, since FRAAs are detected in 58–76% of children with ASD [80] and blood titers of these autoantibodies correlate with folate levels in the cerebrospinal fluid [80,81]. This discovery of the impairment of folate metabolism has recently led to encouraging treatment opportunities [80].

### 1.5. Interplay between Epigenetics and Immune Response in Neurodevelopment

Currently, several studies are addressed to maternal immune activation (MIA) and compounding evidence supports a role for MIA at specific time frames in the pathogenesis of ASD. Several risk factors and pathogenetic pathways may converge and influence fetal brain development through the intra-uterine immune environment. It has been hypothesized that MIA is an effector arm of the epigenetic dysregulation, and a pivotal role of maternal immune response has been suggested for the downstream behavioral phenotypes observed in ASD and other neuropsychiatric disorders in offspring [82,83,84].

Animal studies show a link between MIA and ASD-like outcomes in offspring; correlation is shown with numerous environmental factors, including infections, toxin exposures, maternal stress, and maternal obesity, all of which impact maternal immune response [85]. Therefore, immune activation is a common pathogenetic pathway triggered by numerous infectious agents and environmental factors [86]. Changes in gestational immune environment are correlated with increased risk for neurodevelopmental disorders [87]. This is consistent with the increased risk for ASD and schizophrenia in the offspring of mothers with autoimmunity [88], allergy, asthma [89], maternal acute stress [90], depression [91], and exposure to environmental pollutants [92]—all conditions correlated with the activation of the maternal immune response [66,93].

Animal models made it possible to study the effects of MIA on fetal brain development [83,94,95,96]. DNA hypomethylation and hypermethylation have both been observed in these animal models and in their offspring. Animal models for MIA are obtained by in utero injection of a synthetic double-stranded mimetic of the RNA molecule (polyinosinic:polycytidylic acid-polyI:C), which triggers an immune response of both innate and adaptive immune regulatory mechanisms in the pregnant rodent female, through the activation of TLR3 and subsequent expression of interferon-1 [97]. The offspring of pregnant mice treated with Poly I:C display all the core deficits associated with ASD [98]. Tang et al. used the MIA mouse model and demonstrated significant epigenetic changes in response to polyI:C exposure in utero. Specifically, the study found that differential abnormalities in histone acetylation occurred in the cortex and hippocampus in response to polyI:C exposure; a majority of the observed abnormalities occurred in juvenile mice, prior to the onset of behavioral phenotypes; genes in the glutamate receptor signaling pathway were particularly associated with epigenetic changes in response to prenatal immune activation [94]. MIA was found to dysregulate key aspects of fetal brain gene expression that are highly relevant to the pathophysiology affecting ASD. For instance, transcriptional and translational programs, that are downstream targets of highly ASD-penetrant FMR1 and CHD8 genes, are heavily affected by MIA [83], as well as genes relevant for gamma-aminobutyric acidergic differentiation and signaling and Wnt signaling (that is, a group of signal transduction pathways that regulate crucial aspects of cell fate determination, cell migration, cell polarity, neural patterning, and organogenesis during embryonic development) [96]. Moreover, the changes were markedly influenced by the precise timing of prenatal immune activation, since the early and late gestational windows clearly differed in terms of the altered methylation pattern they induced. Particularly, late prenatal immune activation induced methylation changes in genes critical for GABAergic cell development and functions [96].

Non-coding RNAs, such as microRNAs, represent a further epigenetic mechanism. They resulted in being differentially expressed in brain tissue [40] and in the periphery (i.e., serum/plasma, saliva) [41] in ASD subjects compared to controls. Notably, most differences in ASD patients involved immune response and protein synthesis regulation [99].

Abnormalities in immune response in ASD brain are also reported [96]. Genome-wide transcriptomic studies demonstrate that ASD brains are enriched for “activated” M2 microglial genes and innate immune response-related genes [100]. Nardone et al. determined the presence of many dysregulated CpGs in two cortical regions in brain tissue from people with ASD: Brodmann area 10 (BA10) and Brodmann area 24 (BA24) [101]. Findings in BA10 showed very significant enrichment for genomic areas responsible for immune functions among the hypomethylated CpGs, whereas genes related to synaptic membrane were enriched among hypermethylated CpGs. An inverse correlation links gene expression and DNA methylation. It was reported that genes such as C1Q, C3, ITGB2 (C3R), and TNF-α, important molecules in immune response and implicated also in synaptic pruning and microglial cell specification, are among the hypomethylated (and overexpressed) genes in ASD [101].

Several studies have focused on the association between human leukocyte antigen (HLA) genes and the risk of ASD [102,103]. HLA is a complex genetic region with a pivotal role in some autoimmune diseases and in response towards infection, but also in fetal tolerization. Therefore, HLA impact in ASD may not be ascribed to a single specific HLA gene, but rather to a series of different genes which may intervene at different stages. HLA class I molecules were demonstrated to play a complex and significant role in brain development [104].

HLA class I and HLA-G interact with killer immunoglobulin like receptors (KIR) expressed by natural killer (NK) cells, the effectors of innate immunity. During pregnancy, NK are highly concentrated within the uterine mucosa, at the fetal/maternal interface. NK may be activated or inhibited through the interaction of specific activating and/or inhibitory KIR with HLA-C and non-classical HLA-G molecules on fetal trophoblast. KIR–HLA interaction has been largely demonstrated to play an important role in pregnancy complications. Notably, complications of pregnancy are, together with autoimmunity, very common in ASD mothers [88,105,106,107,108,109].

A skewing of the KIR-HLA complexes, in which activating molecules prevail, was shown in ASD children [110] and their mothers [103]. Moreover, an important role has been suggested for non-classical HLA-G polymorphisms in ASD mothers [111] as well as in women with recurrent spontaneous abortions [112,113]. The generation of a poorly tolerogenic fetal environment results in MIA and may be associated with pregnancy complications as well as with ASD development [111,114]. KIR genes may be regulated by switch on/switch off epigenetic signaling; DNA methylation plays a crucial role in shaping the KIR repertoire, supporting the importance of epigenetic mechanisms as regulatory switches in the immune system early in life [115].

The driving mediators of MIA-associated ASD pathology are most likely elevations in maternal cytokines and chemokines. Therefore, MIA can be considered as a ‘priming condition’ for neurodevelopmental disorders, a susceptibility background on which further risk factors can then be established [116], behaving as ‘multiple hits’ with synergistic effects. These can be infectious agents, any kind of immune stimulation and exposure to toxic substances (particularly, alcohol and drugs) [93]. In MIA murine model, a pro-inflammatory phenotype of T lymphocytes and myeloid cells are common findings [66,87]. Confirming the close relationship and interdependence between the nervous system and the immune system [117], reconstitution of MIA offspring with normal bone marrow improves repetitive behaviors and anxiety, suggesting that some MIA phenotypes are causally related to immune and nervous system imbalances [65].

Elevations of cytokines and chemokines in both maternal serum and amniotic fluid are associated with an increased risk of ASD [118]. Maternal cytokines that cross the placenta, such as IL-6 and IL-4, may alter fetal epigenetic machinery [82]. Particularly, IL-6 and IL-17 may favor inflammation either at the placenta or directly in the developing fetal brain [119]. IL-6 has been identified as a key intermediary of the pathways whereby MIA alters fetal brain development. Maternal IL-6 crossing the placenta can directly affect the development of the fetal brain [120], and animal models show that IL-6 is critical for mediating the behavioral and transcriptional changes in the offspring [121]. Recently, the importance of T helper 17 (Th17) lymphocytes and IL-17a effector cytokine in inducing autism-like phenotypes, acting on the developing fetal brain, has been demonstrated. It is likely that some environmental factors related to the onset of ASD follow this pathway. Structural similarities found between IL-17 family-cytokines and the neurotrophins (proteins regulating survival, development, and functions of neurons) suggest that the IL17Ra pathway has a physiological function in the fetal and adult brain [119].

Maternal cytokines and chemokines, in addition to their activity as immune mediators, are involved in migration of neuronal precursors, neuronal maintenance, synaptic pruning, and neuroplasticity [122]. Sotgiu et al. recently reviewed the numerous abnormalities in immune pathways involved in embryo-fetal neurodevelopment and linked to ASD. The authors confirmed MIA as a predisposing condition for a multiple-step frailty in brain growth, suggesting the importance of care addressed to women before and during pregnancy [123].

Discussion of immune response in pregnancy cannot leave out the microbiota. During pregnancy, one should consider both the placental microbiota and the maternal microbiota. As for the former, for a long time fetus and placenta have been considered to be sterile. Currently, mounting evidence suggests the occurrence of a fetal microbial colonization; moreover, placenta has been reported to harbor a specific microbiota [124]. Even though the ‘sterile womb’ paradigm is debated and results are conflicting, there is convincing evidence that the composition of the maternal microbiota may impact on fetal immune development prior to delivery. The maternal microbiota may exert an indirect effect on the fetus via maternal factors such as maternal immune responses, microbial metabolites that cross the placenta [125,126], or more indirectly via factors that may mediate epigenetic programming in the fetus, such as diet [127] or stress [128], which also affect the maternal microbiota. The gut and vaginal maternal microbiota changes with gestation [129,130]. It is plausible that these changes have an adaptive value. It has been suggested that they allow the fetus to derive energy from the mother’s blood more efficiently [129] and promote immune tolerance in the mother [131]. Studies on animal models suggest that transient changes in maternal microbiota during pregnancy drive fetal immune programming [125].

At present, available knowledge about the fine and delicate tuning of placental immune response and the numerous immune mechanisms impacting fetal neurodevelopment, put the maternal immune response in the spotlight, as one of the most relevant issues to be addressed in the care of pregnant mothers.

### 1.6. Take Home Message from the Interplay between MIA and Epigenetics in Pregnancy

The striking amount of studies showing the impact of maternal immune response on neurodevelopment and the interplay between MIA and the epigenetic machinery requires best efforts to support maternal well-being during pregnancy, as a crucial determinant of lifelong physical and mental health of humans [132]. In fact, the complex molecular machinery described above represents an adaptive response to events occurring during pregnancy, constituting predictable and potentially preventable events.

Numerous risk and protective factors have been demonstrated and linked to the onset of ASD in the offspring, providing suggestions for clinical practice. Emberti Gialloreti et al. recently reviewed risk and protective factors related to the occurrence of ASD in offspring, highlighting the need of care for maternal diet, nutraceutical supplementation, prevention and treatment of metabolic abnormalities, prevention from toxicant exposure, and numerous other factors linked with an increased risk for ASD [133]. Therefore, nutritional state and proper nutraceutical supplementation during pregnancy should be warranted and carefully monitored (enough/not too much). In fact a ‘U shaped’ relationship is reported between maternal multivitamin supplementation frequency and ASD occurrence in offspring [134]. As far the evaluation of risk factors, it is underscored that most exposure models from epidemiological literature may suffer from oversimplification in case the effects of single factors are evaluated separately. In fact, findings in animal models suggest that the study of synergistic–rather than single-effects seems to be correct [135], a concept raising a relevant methodological issue, consistent with the complexity of biological systems. Many studies assessed the frequency of potential environmental risk factors in pregnancy related to ASD in offspring. Grossi et al. highlight explicit associational schemes between risk factors and ASD outcome through a multivariable modeling of data using Auto Contractive Map artificial neural network (ANN). The authors suggest that ANN might highlight hidden trends and associations among the variables, thus revealing the risk profiles related to ASD [136]. Notably, the graph of the study shows that cesarean section, absence of breastfeeding, and early antibiotic use are close to the autism node. All these risk factors are linked to changes in the neonatal bacterial substrate, confirming the importance of the appropriate composition of the early microbial communities for the neurodevelopment [136]. A recent meta-analysis confirms the cesarean delivery as a risk factor for ASD [137]. As a whole, these findings encourage best efforts in favoring vaginal delivery; in case of C-section, the early restoring of a microbial balance should be a priority in primary prevention. In addition to the review of pregnancy risk factors, the methodological question is the most notable feature of the study by Grossi et al. [136]. The study provides relevant suggestions for both research and clinical practice, in order to build an ever-increasing database, which might be continuously fed by clinical, laboratory, and instrumental records. A future personalized application of machine learning systems in neurodevelopmental disorders might be the development of predictive models to track different risk profiles in the lifelong neuropsychiatric trajectory.

### 1.7. Neuroinflammation and Gut-Brain Axis in People with ASD

Findings pointing to cerebral inflammation in autoptic brains of autistic people aroused considerable interest pertaining to the involvement of the immune response in ASD [138]. The study showed neuroinflammatory activity in the cerebral cortex, white substance and cerebellum of autistic subjects. Furthermore, marked activation of microglia and astroglia was demonstrated [138].

More recently, growing evidence supported a role for dysregulated neuroinflammation. In a recent review by Matta et al., numerous studies are reported concerning reactive microglia and astrocytes, altered glial structure and function, cytokine profiles, and gut immune dysfunction in ASD people and animal models. The authors conclude that a strong evidence for nervous system interaction with immune pathways in ASD is demonstrated [139].

The association between ASD and immunological imbalance concerns both innate immunity and the specific response of T lymphocytes, with a shift towards a Th1-type pattern and prevalence of pro-inflammatory cytokines; increase of B lymphocytes, NK cells, and dendritic cells; and different patterns in the expression of surface markers [102]. Findings in brain tissue from individuals with ASD do not allow to establish the period in which neuroinflammation started and the ways whereby it interfered with the formation of the neural networks. Another open question concerns the sequence of events. It is not clear to what extent neuroinflammation is a contributory effector mechanism of neurodevelopmental disorder or rather whether immunological abnormalities are secondary to a systemic and complex biochemical/metabolic imbalance. In other words, to what extent neuroinflammation in ASD is primary (causal) or secondary (reactive)? As the immune system is primarily involved in tissue repair and homeostatic processes, immune findings in ASD could represent compensatory responses to dysfunctional network activities and cellular stress. Therefore, studies addressing the temporal dynamics of brain dysfunction with age and whether they are linked to the ongoing and dynamic immune changes are important areas for future research.

Taken together, findings about immune abnormalities support evidence of an early and ongoing dysfunction in the peripheral immune system and the brain of individuals with ASD [140].

Severe immune alterations are demonstrated also in the non-ASD siblings of ASD patients [141]. In particular, in both autistic children and their siblings, increase in the production of IL-10 and a skewing toward earlier, less differentiated lymphocyte subpopulations were showed. Notably, IL-10 has strong anti-inflammatory properties: This finding could thus be interpreted as a way whereby the immune system tries to counterbalance the inflammation present both in autistic patients and in their unaffected siblings [141].

Neuroinflammation in ASD shows crucial links with the gut–brain axis, a bidirectional neurohumoral communication system [142] orchestrated by the microbiota [143].

Research on gut–brain interactions in the last decades has provided evidence about the close interactions between the gut-associated immune system, enteric nervous system, and gut endocrine system [144]. Animal models and studies in humans seem to support a relationship between the gut microbiota and brain development; moreover, functions and studies on microbiomes have triggered great interest from professionals and the National Institute of Mental Health. The simultaneous presence of oxidative stress, mitochondrial dysfunction, and inflammation has been often observed in the brain of ASD people, which were correlated also with ASD symptoms, thus suggesting an inter-relationship between these anomalies [145].

### 1.8. Microbiome in the Crosstalk between Immune System, Gut, and Brain

A rapidly increasing amount of evidence point to host–microbe interactions at virtually all levels of complexity, ranging from direct cell-to-cell communication to extensive systemic signaling, involving various organs and systems, and starting even before conception. The traditional idea of an auxiliary function of the friendly ‘intestinal flora? has shifted in the recent years to the assignment of the role of orchestra conductor in the psycho-neuro-endocrine system. Consistently, a growing number of research projects have been launched worldwide concerning this topic. The human microbiota consists of the 10–100 trillion symbiotic microbial cells harbored by each person. The set of genes of the microbiota is collectively known as the microbiome [146], encoding for at least 100 times more genes than our genome [147] and suggesting the’question ‘who is harboring whom?’. As a whole, each individual should be considered an “holobiont harboring an hologenome’. The GI tract is the most heavily inhabited organ with micro-organisms, harboring a huge diversity with more than 500 bacterial species. [147].

Dysbiosis—the state of unbalanced microbial communities—and its impact on the early shaping of the immune system has been demonstrated in the pathogenesis of a wide range of diseases [148,149,150], including neurodevelopmental and psychiatric conditions [151,152].

Colonization of the infant’s gut represents the de novo assembly of a microbial community [153]. The infant’s gut microbiota is established after birth, within the first three years. After childbirth, the neonate and microbiota develop in an orchestrated way. There is a strong influence on infant microbiota of maternal gut microbiota during pregnancy. Maternal gut strains have been shown to be more persistent in the infant gut and ecologically better adapted compared to those from other sources [154].

The early establishment of gut microbiota is affected by several factors such as delivery mode (cesarean delivery vs. vaginal delivery), breast milk vs. formula feeding, antibiotic usage, timing of the introduction of solid foods, and cessation of milk feeding [155].

Early life perturbations of the developing gut microbiota can impact neurodevelopment and potentially lead to adverse mental health outcomes later in life. Borre et al. compare the parallel early development of the intestinal microbiota and the nervous system. The concept of parallel and interacting microbial-neural critical windows opens new avenues for developing novel microbiota-based preventive and therapeutic interventions in early life [156].

Animal studies suggest that the microbiota may regulate microglia maturation and function by activating immune signaling pathways, the release of cytokines, and other inflammatory molecules [102,140,157], including inflammasomes activation [158].

Other mechanisms are involved in the communication between gut microbiota and the brain and have been proposed to explain the possible role of microbiota in neurodevelopmental disorders: direct activation of the vagus nerve [151]; production or alteration of neurotransmitters, including serotonin [151]; production of toxins [159]; aberrations in fermentation processes or products [160,161]; and dysbiosis-induced breakdown in gut integrity [162,163]. Interacting molecules may be produced by the gut microbiota, such as short-chain fatty acids (SCFA), which may cause the increase of gut permeability and then act on a range of other systems. SCFA may also affect epigenetic modifications. In particular, butyrate is a histone deacetylase inhibitor, contributing to the attainment of a less relaxed chromatin conformation. This small molecule can cross the blood–brain barrier and impacts epigenetic machineries in the brain [164]. Butyrate exerts anti-inflammatory and neuroprotective effects [165] and attenuates social behavior deficits in autism models [166]. It supports mitochondrial function, stimulating oxidative phosphorylation and fatty acid oxidation [167]. Its concentration has been shown to be reduced in fecal samples from ASD children [168], and its supplementation had a positive effect in lymphoblastoid cell lines derived from children with ASD under physiological stress, and, in particular, in cell lines with underlying mitochondrial dysfunction [169], providing interesting insight into links between ASD, mitochondria, and gut microbial communities and the possible clinical application. Microbiota may also mediate the availability of S-methyl-Methionine (SAM), the donor of methyl groups for DNA methylation (reviewed by Kaur et al., [170]) by producing folate for generation of SAM. Folate is generally obtained by appropriate diet. Notably, a key enzyme for regulating the availability of folate for either DNA synthesis or DNA methylation is methylenetetrahydrofolate reductase (MTHFR). Remarkably, some MTHFR genetic polymorphisms have been associated with ASD risk [134,171]. This is another example of how inadequate diet or gut dysbiosis may mimic genetic defects promoting the onset of ASD.

In addition, stress, as signaled via the hypothalamic-pituitary-adrenal (HPA) axis, is one top-down mechanism that may affect gut microbiota [172]. As far as the composition of microbiota in ASD subjects, conflicting results are reported by numerous studies. A shift in the microbiota in autistic individuals compared with controls was reported and included elevated Clostridia spp., Bacteriodetes, and Desulfovibrio spp. in ASD [160,163,173,174]. The complexity of community relationships within the microbiota and the current challenges on microbiota data analysis—risk of false positive discovery—might explain the wide variability of findings [175].

In addition to dysbiosis, compared to controls, ASD patients show an increase in gut permeability (the so-called ‘leaky gut’), a finding supported by numerous studies showing alterations in gut barrier in ASD [176]. Among factors concurring to the diagnosis of gut permeability, loss of zonulin is one of the most important. As a ‘biological door to inflammation’ [177], higher levels of plasmatic zonulin are reported in people with ASD than controls [178].

Fecal calprotectin might be a useful biomarker in the assessment of gut–brain axis involvement. It identifies people with gut inflammation [179] and correlation between calprotectin levels and main domains of the autism diagnostic interview-revised (ADI-R) has been shown [180].

In addition to theoretical issues and laboratory data, clinical findings also support the hypothesis of a pivotal role of the gut–brain axis, immune activation, and microbiota in ASD. A high rate of allergy and gastrointestinal (GI) symptoms are reported in people with ASD [2]. Diarrhea, constipation, vomiting, reflux, abdominal pain/discomfort, flatus, and unusually foul-smelling stools are more frequent than in healthy controls [181,182]. In a large sample of adult ASD patients, GI complaints are reported in 21% of patients [183]. A meta-analysis from ‘Pediatrics’ confirmed a higher prevalence of GI symptoms among children with ASD compared with control children [12].

Abnormalities in GI motility and intestinal permeability have been reported [175]. Studies report a wide range of variability of GI symptoms, from 9 to 70% [184]. Differences in studied populations and different assessment tools for symptoms might explain these differences. What is not in dispute is that GI disturbances represent a topical issue among health needs for people with ASD, with severe impacts on wellbeing and variably contributing to behavioral abnormalities [182,184]. Difficulties in the recognition of pain in people with intellectual disability make it easy to underdiagnose pain and discomfort. The risk of underestimation is even higher in ASD, due to difficulties in communication and abnormalities in the neural integration of somatosensory afferent inputs. For this reason, proper tools for GI symptoms and pain evaluation should be systematically included in clinical assessment and parallel monitoring of behavioral symptoms and of any other ailment should be provided [185]. In fact, inputs from the internal environment, as well the influence of external environmental factors, represent antecedent events preceding behaviors, and requiring evaluation according to the behavioral functional analysis methodology [186]. Therefore, a preliminary medical assessment for the identification and treatment of pathophysiological comorbidities of ASD is expected to achieve optimal outcomes according to a multidisciplinary approach [187,188].

Neuroinflammation in ASD might have links with epilepsy [189]. In a population-register study, a quarter of children with ASD had epilepsy, in contrast to 1.5% of population-based controls [190]. Berg et al. report a prevalence of 7% among children with no motor deficits or severe intellectual disability, compared to 42% in people with motor deficits and severe intellectual disability [191]. The above-mentioned evidence of immunologic abnormalities in ASD suggests a possible role of neuroinflammation in the pathogenesis of epilepsy [192]. Inflammatory mediators, such as IL-1β (imterleukin-1 β), TNF (tumor necrosis factor), HMGB1 (high mobility group box 1), TGF- β (transforming growth factor-β), and prostaglandins, can alter neuronal, glial, and blood–brain barrier functions by activating transcriptional and post-translational mechanisms in brain cells. Furthermore, a role by brain mast cells in neuroinflammation is reported, and the involvement of these cells is hypothesized in the pathogenesis of epilepsy in a group of ASD people [192]. The impact of immune abnormalities on the occurrence of epilepsy in ASD is of utmost interest and deserves further study.

In summary, clinical findings confirm the pivotal role of immune abnormalities and the gut–brain axis in ASD. Therefore, expertise in medical assessment for comorbidities should be warranted.

### 1.9. Mitochondria/Oxidative Stress … and the ‘Bad Trio’

Increasing and converging evidence suggests a pivotal role for mitochondria in neurodevelopmental disorders [193]. Beside the generation of energy in the form of ATP, mitochondria encompass a wide array of functions, ranging from metabolite and redox signaling to the regulation of nuclear gene expression and epigenetics [194,195,196].

Notably, energy provided by mitochondria oils the epigenetic machinery, allowing selective access to specific DNA sequences by regulating the various levels of chromatin structure, from nucleosomes to chromatin fibers [197]. In neurodevelopment, mitochondria emerged as key regulators of neural stem cell fate decisions, impacting neurogenesis both in neurodevelopment and in adult mature brains [198].

Many studies have shown that mitochondrial dysfunction contributes to placental pathology underpinning gestational disorders [199]. Mitochondria are sensitive stress targets in the placental microenvironment. Placenta development and a successful pregnancy are under a precise oxygen-dependent control of trophoblast migration/invasion [200] and maternal immunity [201], since a regulatory loop might exist between trophoblasts and maternal immune cell subsets, promoting the harmonious maternal–fetal crosstalk [202]. Persistent low oxygen pressure, leading to failed trophoblast invasion, promotes inadequate spiral artery remodeling, a characteristic of preeclampsia [200].

Mitochondrial dysfunction and oxidative stress are two major and interconnected metabolic abnormalities associated with ASD, since oxidative stress causes mitochondrial dysfunction and dysfunctional mitochondria produce Reactive Oxygen Species (ROS) [203]. Thus, mitochondrial dysfunction can be at the same time the cause and/or the result of oxidative stress. In fact, excessive free radical production can lead to mitochondrial damage, and, in turn, the damaged mitochondria are prone to release increased amounts of ROS; this process is maximized in what has been termed ‘Ros-induced ROS release’ [204], but is also a common evidence in pathologies characterized by chronic oxidative stress. Under this perspective, the primary source of mitochondrial dysfunction may be oxidative stress itself, which in turn may originate from manifold ROS-generating processes, including chronic inflammation [205], metabolic dysfunctions [206,207], exposure to heavy metals [208], and other environmental issues. Indeed, several environmental factors, including toxicants, microbiome metabolites, and an oxidized microenvironment are shown to modulate mitochondrial function in ASD tissues [203,209]. Both intrinsic and extrinsic stressors can impact the interplay by increasing ROS and/or reducing mitochondrial function, thus prompting the establishment of a vicious circle [203,210]. Numerous genetic abnormalities are associated with mitochondrial dysfunction in ASD [210,211,212]. Furthermore, several environmental factors, including toxicants, microbiome metabolites, and an oxidized microenvironment are shown to modulate mitochondrial function in ASD tissues [203,209].

Traditional biomarkers commonly used to identify mitochondrial dysfunction include lactate, pyruvate, alanine, and creatine kinase. A meta-analysis by Rossignol and Frye demonstrated that ASD was associated with higher levels in lactate, pyruvate, lactate-to-pyruvate ratio, alanine, creatine kinase, ammonia, and aspartate aminotransferase (AST), and in decreased carnitine concentration [210,211]. Among mitochondrial dysfunctions, abnormal activity of the electron transport chain (ETC) enzyme complexes—the machinery fueling energy production—is reported in ASD children. Notably, these abnormalities are found in mucosal samples taken both from rectum and caecum and might explain gut dysmotility, higher sensitivity to oxidative stress, and abnormal functioning of enterocytes [213]. Oxidative stress results in damaged proteins and lipids in the cell, and consequently impacts enterocyte function. Therefore, dysbiosis (that is imbalance in microbial metabolites) and oxidative stress might explain abnormal mitochondrial function in the caecum [213]. Additionally, gut dysmotility caused by mitochondrial dysfunction would explain constipation observed in ASD and other GI symptoms [210,211]. Interestingly, GI problems similar to those of autistic children have also been noted in children affected by genetic syndromes in which mitochondrial dysfunctions play a central role in the etiopathogenesis and having ASD among clinical features, such as Down syndrome and Rett syndrome [210,211,214], again suggesting a link between mitochondrial dysfunction, GI problems, and microbiota in ASD people.

Numerous studies have reported biomarkers representing abnormalities in fatty acid metabolism in ASD [215]. ASD patients from Saudi Arabia were found to have elevations in saturated fatty acids and depressions in polyunsaturated fatty acids as compared to age-matched controls [216], consistently with previous results suggesting polyunsaturated fatty acids, carnitine, and lactate as biomarkers of brain energy in children with ASD [217].

Children with ASD show low levels of the reduced form of glutathione (GSH), the major intracellular antioxidant responsible for maintaining redox homeostasis and for reducing ROS in the cytosol and mitochondria [215]. In addition, more than 30% of ASD patients have elevations in acyl-carnitine, a cofactor carrying long-chain and very-long-chain fatty-acids into the mitochondria. Interestingly, this same pattern of GSH and acyl-carnitine abnormalities found in children with ASD [218] was also found in the rodent propionic acid (PPA) model of ASD [219]—PPA being one of the most important microbial metabolites believed to cause systematic mitochondrial dysfunction [218]—thus providing further evidence for the association among PPA, mitochondrial dysfunction, and ASD [220].

This is consistent with finding of higher levels of PPA in fecal microbiota and metabolome of children with ASD [161]. Balance in microbial metabolites (enough/not too much) significantly impacts mitochondrial functions and influences GI activity. For example, butyrate is converted into acetyl-CoA, which then is utilized in the citric cycle for NADH production. NADH, on the other hand, is utilized by the mitochondrial ETC complex I, the main site of entrance of reducing equivalents into the ETC, crucial for respiration and energy production, and the main site of ROS production when it is dysfunctional [213].

Therefore, at least part of the effects of dysbiosis on neurodevelopment and GI involvement in ASD seems to be mediated by mitochondrial impairment [210,211,221].

There are at least three possible connections between the GI tract and mitochondrial abnormalities in ASD [222]. First, mitochondrial dysfunction itself could result in GI dysfunction [223]. Secondly, there are common exposures to environmental stressors that are associated with ASD that can affect both the mitochondria and the GI tract: pesticides and heavy metals [224], exposure to drugs such as acetaminophen [225] or antibiotics, either during pregnancy [226] or early in life [227,228,229], and more likely the exposure to all these and other factors taken together [230]. Another plausible connection between gut and mitochondrial impairment is represented by cell wall agents (i.e., lipopolysaccharide, [231]) or metabolites from enteric bacteria [160,221] and their effect on mitochondrial functions.

Among bacterial metabolites, the aforesaid propionate (PPA) is seemingly the Short Chain Fatty Acid mostly produced by micro-organisms prevalent in the gut of ASD patients, including Clostridia spp., Bacteriodetes, and Desulfovibrio spp. in ASD [163,173,174,232]. Furthermore, propionate is universally used as a preservative in processed food due to its anti-fungal characteristics [233].

Maternal PPA exposure is one of the possible mechanisms interfering with neural wiring during early stages of embryonic neural development and leading to a shift of glial cells towards an inflammatory pattern [233]. Notably, the exposition of human fetal-derived neural stem cells to PPA resulted in downregulation of PTEN expression and a consequent differentiation shift to gliosis and neuroinflammation [233].

Another important point involving fatty acids is the organizational and functional integrity of the cellular membrane. The membrane phospholipids—the building blocks of membranes—are characterized by a balance in the diverse fatty acid residues (saturated, monounsaturated, and polyunsaturated), which varies from tissue to tissue in the same body [234], and is a condition-sine-qua-non for the normal health of the cells. An inadequate dietary intake, poor availability of specific enzymes, and oxidative stress alter the membrane lipids and the functionality of embedded proteins. Indeed, impairment in function of erythrocyte membrane proteins and lipids have been demonstrated as consequences of increased oxidative stress in ASD. A very significant reduction of Na+/K+-ATPase activity (−66%, *p* < 0.0001), a reduction of erythrocyte membrane fluidity, and alteration in erythrocyte fatty acid membrane profile (increase in monounsaturated fatty acids, decrease in EPA and DHA-ω3 with a consequent increase in ω6/ω3 ratio) were found in ASD children compared to controls [235]. Interestingly, some clinical features of children with ASD (in particular, hyperactivity and cognitive development) showed correlation with some parameters of the lipidomic profile (saturated fatty acids, arachidonic acid) and membrane fluidity, highlighting a pathogenetic key-point in ASD and a potential use of membrane lipidome profile as useful biomarker for personalized therapeutic supplementation [236]. The importance of a correct membrane concentration of DHA-ω3 was confirmed in a study of membrane lipidome, showing that the decrease of this fatty acid is not attributable to dietary differences between healthy and diseased children, as evaluated by food questionnaire indicating, for example, fish consumption. Moreover, statistical significance test of the ROC curve for DHA (*p* value = 0.0424) with a cut-off value at 4.08% gave a significant odds ratio corresponding to 6.23 (*p* value = 0.017; IC 95%: [1.3956–27.8412]), indicating that individuals with values of DHA < 4.08% (cut-off) have a probability of being autistic 6.23 times higher than those with DHA > 4.08%. [237]. A correlation between the reduced membrane fluidity and striking morphological abnormalities in the shape of red blood cells was also demonstrated [238], where most of the biological alterations resulted to be ascribed to oxidative stress [238]. As noticed by the authors, findings suggest a plausible dysfunction of erythrocytes in tissue oxygenation [238]. If so, a chronic state of hypoxia in tissues is expected to worsen the oxidative stress, contributing to a vicious loop and ongoing deterioration of health in ASD people.

A relevant increase in oxidative damage markers was further confirmed by protein glycation, oxidation, and nitration adducts and amino acid metabolome in plasma and urine of children with ASD [239]. Findings in people with ASD could be well described by the striking definition of ‘pervasive oxidative stress’. Rossignol and Frye reviewed interplay between oxidative stress and immune activation [210,211]. The increase in the gene expression of IL6 and the stress protein HSP70i was demonstrated in ASD children [240]. Furthermore, the study of the protein expression of the antioxidant enzyme family of peroxiredoxins showed a significant increase in plasma of ASD children, supporting the link between oxidative stress and neuroinflammation in ASD [240]. The interplay between mitochondria and immune response represent a complex bidirectional system involving numerous mechanisms. Metabolic pathways such as tricarboxylic acid cycle, oxidative phosphorylation, and fatty acid oxidation impact macrophage polarization and T cell differentiation; mitochondrial ROS control immune cell transcription, metabolism, and NLRP3-mediated inflammation; mitochondrial DNA can be released from mitochondria into the cytosol and activate the NLRP3 inflammasome and production of IL-1β and IL-18 [241]. Findings are consistent with the aforesaid immune abnormalities in ASD [140], including higher inflammasome activation (in particular, NLRP3 activation) than in controls [158].

In summary, the literature findings reported above suggest in ASD the existence of a vicious circle between dysbiosis, immune response, and mitochondrial dysfunction/oxidative stress, a ‘bad trio’ which might start from the embryo-fetal period, impact neurodevelopment, and even might cause a progressive worsening of the neurological disorder. In fact, the same ‘bad trio’, if not stopped, might go on and contribute to the worsening of the systemic disorder through all life.

The interplay between the main effector pathways causing the ASD phenotype and acting during the embryo-fetal stage all through life is illustrated in Figure 1.

### 1.10. Metabolomics: A Promising ‘Meaningful Web’ Describing a Biochemical Fingerprint

The evolving spectrum of clinical presentation and of laboratory findings in ASD offers the challenge to understand and respond to similarly evolving health needs of a growing number of people. Therefore, ASD is a paradigmatic situation urgently requiring a dynamic and personalized approach. The availability of suitable diagnostic tools capable of grasping the biological complexity seems to be the starting point.

Currently, sensitive, specific and early biomarkers are not available to detect ASD before the clinical onset of behavioral abnormalities; therefore, professionals have at their disposal only standardized clinical tools—interviews and behavioral scales—to make diagnosis. In the complex biological scenario beyond behavior in ASD, so distant from a linear model of ‘a symptom, a biomarker’, metabolomics opens new interesting avenues. In fact, it describes the individual molecular phenotype and allows monitoring of its changes over time.

The molecular phenotype closely reflects the result of interplay between genomics, transcriptomics, proteomics, environmental factors, and gut microbiota [242], and might thus be associated with the type and degree of the behavioral/cognitive impairment and with functional neuroimaging [243].

Metabolomic approach represents the phenotype by the detection and the representation of metabolites, low-molecular-weight end-products of cellular metabolic pathways, which in turn are influenced by genetic and non­genetic factors. Metabolomics allows the systematic identification and quantification of the global collection of all metabolites, namely the metabolome, recognizable either in biological fluids (e.g., urine) or in tissues [244]. Metabolites can be identified and characterized in their elemental composition, molecular charge and mass, stereochemical orientation, and order of atoms [245]. Metabolomics accurately identifies metabolites involved in the same pathway as well as the metabolic network shaped by nodes (metabolites) and their interactions (scale-free network models) [246]. In other words, metabolomics provides a personalized description through a ‘meaningful web’, representing the individual biochemical fingerprint.

Today, high throughput technologies like proton nuclear magnetic resonance (1H NMR) spectroscopy, liquid chromatography, and gas chromatography coupled with mass spectrometry (LC­MS and GC­MS, respectively) and further sophisticated analytical methods are outstanding tools that allow researchers to accurately explore the metabolome and its variations over time in various perinatal conditions involved in ASD etiology, for example perturbations of the gut–brain axis, due to gut dysbiosis, increased intestinal permeability, inflammation, oxidative stress/mitochondrial dysfunction, well representing the ‘juniper bush’ of ASD [242].

This means a great opportunity to search for new highly sensitive and specific biomarkers for early diagnosis of ASD, risk of regressive ASD [247], and further disorders in neurodevelopment and psychopathology, up to adulthood [248,249]. In a similar way to that of other neuropsychiatric disorders, ASD may be closely associated with several maternal, fetal, and perinatal epigenetic factors that influence brain development and maturation [250,251]. Metabolomics allows the discovery of biomarkers for an early diagnosis and the monitoring of fetal and perinatal programming [252]. The detection of biochemical patterns suggestive for the vicious circle within the ‘bad trio’—involving maternal dysbiosis, immune activation, and oxidative stress—could allow early and personalized interventions during pregnancy, with the possibility to closely monitor the effects of treatment through changes in metabolomic profile. The urinary metabolome of ASD children has been extensively studied, and some studies have been devoted also to the analysis of the plasma metabolome. Mussap et al. reviewed most relevant metabolic pathways and key metabolites implicated in ASD. The most discriminant metabolites in ASD were involved in amino acid metabolism, antioxidant status, nicotinic acid metabolism, and mitochondrial function [253]. Most of the studies in ASD reported abnormalities in gut bacterial-derived compounds and in intermediary compounds of the Krebs cycle [254,255,256], confirming the aforesaid pivotal role of oxidative stress, microbiota, and abnormalities in mitochondrial function in ASD [203,210,211,221].

In summary, metabolomics approach opens very promising perspectives in diagnosis and follow-up in ASD, allowing an early understanding of the individual ASD patient, with evolving and unique needs.

## 2. New Methods for Renewed Diagnostic Tools: Machine Learning System in EEG

ASD is associated with abnormal neural connectivity [257,258,259,260,261,262], and some abnormalities in brain development might be already detectable at birth.

Currently, neural connectivity is a theoretical construct that is hard to be measured, but research in network science and time series analysis suggests that the neural network structure—a marker of neural activity—is measurable by EEG [263].

Hustler et al. described three types of cortical construction abnormalities in ASD (a) alterations to columnar structure that have significant implications for the organization of cortical circuits and connectivity; (b) alterations to synaptic spines on individual cortical units that may underlie specific types of connectional changes; and (c) alterations within the cortical sub-plate—a region that plays a role in proper cortical development and in regulating interregional communication in the mature brain [264]. The relevant involvement of the cerebral cortex in the substantial alteration of the cortical circuitry explains the unique pattern of deficits and strengths that characterize cognitive function. These findings make electroencephalography (EEG) a plausible useful tool to detect these abnormalities.

The EEG can measure neural activity and may provide a useful tool to early detect ASD in children, thus allowing the opportunity for early intervention. The potential usefulness of EEG in ASD has been reviewed almost ten years ago [259], in order to examine evidence for the utility of three methods of EEG signal analysis in the ASD diagnosis and subtype delineation. All studies identified significant differences between ASD and non-ASD subjects, confirming the presence of specific EEG abnormalities. However, due to the high heterogeneity in the results, findings could not be generalized and none of the methods, if taken alone, has been proposed as a new diagnostic tool [259].

Recent studies on this topic open new avenues and might represent a turning point for the early diagnosis of ASD based on the analysis of electroencephalographic tracing (EEG) supported by new adaptive artificial systems (ANNs). It was hypothesized that the atypical organization of the cerebral cortex in ASD might translate into an EEG signature detectable through powerful analytical systems such as ANNs [265,266,267].

Using particularly advanced machine learning systems, it has been possible to build a software able to distinguish almost perfectly the EEG from subjects with ASD from those of neurotypical controls or with different neuropsychiatric disorders.

The new system, called MS-ROM/I-FAST, belongs to the family of systems developed by the Semeion Research Centre. MS-ROM/I-FAST is a new and complex algorithm for the blind classification of the original EEG trace of each subject, through the recording and analysis of a few minutes of their EEG without any preliminary pre-processing [266]. A first pilot study assessed the discriminatory power of the methodology in distinguishing subjects with ASD from neurotypical controls. After the MS-ROM/I-FAST pre-processing, the overall predictive capacity of the different automatic learning systems in distinguishing autistic cases from the controls was constantly 100% [266]. Notably, these results were obtained at different times and in separate experiments performed on the same training and testing subsets. Furthermore, the similarities between the weight matrices of the neural networks measured with appropriate algorithms were not influenced by the age of the subjects, suggesting that the networks read invariant characteristics related to the disconnection signature in the brain [266]. The results of the pilot study have been recently confirmed. EEG data from ASD children were compared with EEG from controls affected by other neuropsychiatric disorders. With the training-testing protocol, the overall predictive capacity of the machine learning system used to distinguish between ASD and controls was constantly over 90% [267]. Along this research area, it would be of utmost interest to extend EEG tracks recording within the first year of life, with the purpose to use this technique as a specific, sensitive, non-invasive, non-expensive tool for early detection of the signature predictive for ASD. The potential usefulness of this methodology might be extended to find out possible different EEG signature in ASD subgroups with different onset (early/regressive autism) and different phenotypes. Furthermore, this tool could monitor the evolution of EEG abnormalities, find hidden links with clinical and laboratory biomarkers and monitor the effect of therapeutic interventions.

## 3. Discussion

Big data from basic research performed over the last ten years need to be translated into clinical practice. Knowledge about the increasing complexity in the etiopathogenetic pathways of diseases is the premise for the suitable adaptation of strategies for prevention, diagnosis, and treatment according to the evolving health needs of the population. In neurodevelopmental disorders—in particular in ASD—most current statistical methods do not seem suitable to study not linear, complex, and fuzzy interactions involving genome, epigenome, environmental factors, and nervous–immune-endocrine interplay, and to do so along a pathway that starts even before conception.

Most of available studies have been designed on the basis of methods developed in the first half of the past century, when the scenario was dominated by acute infectious diseases and linear models apparently succeeded in describing the phenomenon. In the last century, the epidemiological scenario has profoundly and dramatically changed, and traditional methods seem to be able to assess only a very small part of the phenomena, if compared to their intrinsic complexity. Consequently, the development of methods consistent with the complexity of the phenomena seems to be the premise for personalized medicine, able to avoid the narrow view of what is well known, leaving out the broader horizon of the unknown.

ASD is a paradigmatic condition within the epidemiological transition occurring in the last decades toward the prevalence of non-communicable disorders and diseases, which requires a plausible pathogenetic mechanism able to explain both epidemiological and clinical findings, that is the combination of the striking increase in prevalence with the multifaceted phenotype. The need for a scientific consensus on a comprehensive paradigm is much more than a theoretical issue. In fact, the coherent translation of the pathogenetic model into clinical practice is the premise for effective preventive strategies and comprehensive answers to the complex health needs of ASD people.

In order to do so, a dynamic and systemic perspective—starting with the care for women’s health before pregnancy occurs—seems to be the most promising approach to face this major public health issue, both for current needs and in the future perspective.

Embryo-fetal brain development is profoundly influenced by numerous interacting environmental factors, named ‘exposome’ as a whole. Both in intrauterine and in postnatal life, environmental information converges on three major interacting/overlapping pathways: dysbiosis, mitochondrial impairment/oxidative stress, and immune activation (named MIA during pregnancy). As a whole, the three above-mentioned effector pathways—as a pathogenetic trio-impact epigenetic machinery. The matrilinear transmission of both microbiota and mitochondria [268,269] further enforces the need for effective women’s health programs, which are even more important in the presence of known risk factors for ASD, such as of the occurrence of neurodevelopmental disorders in previous offspring.

Prenatal factors are expected to influence development more than all others, and are not limited to brain alone. The multifaceted phenotype and endophenotype found in ASD people are consistent with a multisystemic and evolving disorder. In fact, metabolomic data concerning the ‘bad trio’ are representative of a systemic and evolving inflammatory syndrome. Findings seem consistent with the high prevalence of obesity in ASD and obesity-related disorders (type 2 diabetes mellitus, hypertension, hyperlipidemia, and nonalcoholic fatty liver disease/nonalcoholic steatohepatitis) [270] and of metabolic syndrome in psychiatric disorders [271]. The issue is of the utmost importance and presents fundamental healthcare issues. Among the environmental factors, diet is in the spotlight as a fundamental tool for prevention and care in ASD. In particular, considering the frequent eating disorders [272] and use of edible reinforcers in educational intervention [273], the risk of nutritional imbalance seems to be high in people with ASD and could—at least in part-explain findings consistent with metabolic syndrome and oxidative stress [206,207,270]. Therefore, converging evidence suggests to include nutritional experts in the panel of professionals in the healthcare model addressed to people with ASD. In fact, besides the energy intake, diet impacts microbiota [274,275], immune function [276], and lipidic cell membrane profile [277]. In other words, diet impacts most of the fundamental pathogenetic mechanisms demonstrated in ASD. Consistently, an individualized and monitored dietetic plan may play a central role in preventive strategies and care in ASD.

The proposal of a personalized nutrition plan is only an example aimed at glimpsing the value of interdisciplinary models for clinical cooperation. Suitable diagnostic and monitoring tools are required to grasp the whole complexity of ASD and translate it into concise information, easy to be used by clinicians. Currently, metabolomics and machine learning systems seem to be respectively the ‘materials and methods’ of a foreseen tremendous impact both in research and clinical practice in the field of ASD.

The dynamic trajectory of individual brain connectome and the ‘multiple-hits’ frailty encourage best efforts to attain the early detection of any biological abnormality potentially impacting neurodevelopment, in order to restore the best balance as soon as possible, hopefully in the period of maximum neuroplasticity. Waiting for the availability of metabolomics in clinical practice in the next years, the question arises as to how to start transferring current biological knowledge into medical advice as soon as possible. The involvement and relevance of the gut–brain axis, dysbiosis, increase in intestinal permeability, and abnormal lipidic composition in cell membranes in ASD provide some useful suggestions for the adaptation of clinical assessment. Biomarkers such as fecal calprotectin, zonulin, erythrocyte fat profile, analysis of the microbiota, and of fecal microbial metabolites (mainly, lactate, propionic acid, and butyrate) characterize subgroups of people requiring specific diagnostic and therapeutic interventions addressed to expected and easily testable organic needs. The inclusion of such biomarkers in clinical trials is expected to contribute to the proper evaluation of the effectiveness of interventions on behavioral outcomes.

## 4. Conclusions

Current hardships experienced by autistic people and by their families, and the expected worsening of their troubles in the coming years are telling all of us that it is not time to rest on laurels.

Perhaps it is time to stop a while and take stock of the situation, in order to prevent the plethora of data by the literature that might take the scientific community away from people’s needs instead of match them.

Therefore, ‘joining the dots’ seems to be the premise for a comprehensive and effective healthcare model addressed to ASD people. A multidisciplinary approach and interdisciplinary sharing of knowledge seem to be the only way to answer their complex, evolving, and unique needs. Figure 2 suggests an interdisciplinary healthcare model that is coherent with the contents of this review and comprehensive of their translation into clinical practice.

The author panel proposing the review well represents the ‘spectrum’ of expertise required for advice in the evaluation of ASD patients. In other words, the heterogeneity of their expertise represents the implementation of the need for developing ‘skills in communication and social interaction‘ that is the intriguing challenge that ASD is posing to all of us.

## Figures and Tables

**Figure 1 jpm-11-00070-f001:**
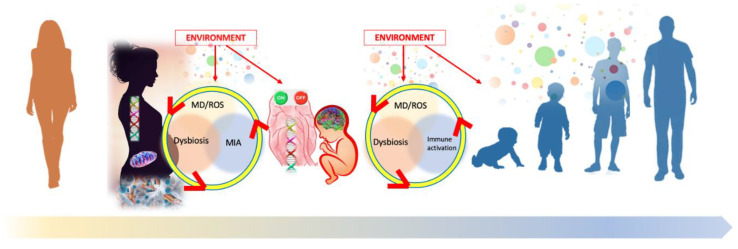
Interplay between the main determinants of Autism Spectrum Disorder. MD, Mitochondrial dysfunction; MIA, Maternal Immune Activation; ROS, Reactive Oxygen Species. In the new individual, matrilinear transfer of mitochondria and microbiota adds to the genetic information stored in the maternal and paternal germinal cells. Environmental factors as a whole may directly affect the epigenetic machinery, as it happens with heavy metals, or may influence the interconnected molecular pathways involved in the ‘bad trio’ (mitochondrial dysfunction (MT)/oxidative stress (ROS) plus maternal immune activation (MIA) plus dysbiosis). The ‘omniscient placenta’ [51] drives the metabolic and epigenetic regulation of fetal programming, hence influencing the ontogenesis and the crucial early stages of neurodevelopment. The epigenome—similarly to a software switching genes on and off [29]—is programmed in an adaptive and predictive sense by the intrauterine and cellular microenvironment, setting the limits of physiological adaptations to the postnatal environment and influencing the lifelong risk for diseases [43]. After birth, the same mechanisms involving environmental factors and the ‘bad trio’ are at play, and may continue to undermine human health lifelong. As for neurodevelopment, the maximum impact occurs in the first two years of life, which is the crucial time window for brain wiring.

**Figure 2 jpm-11-00070-f002:**
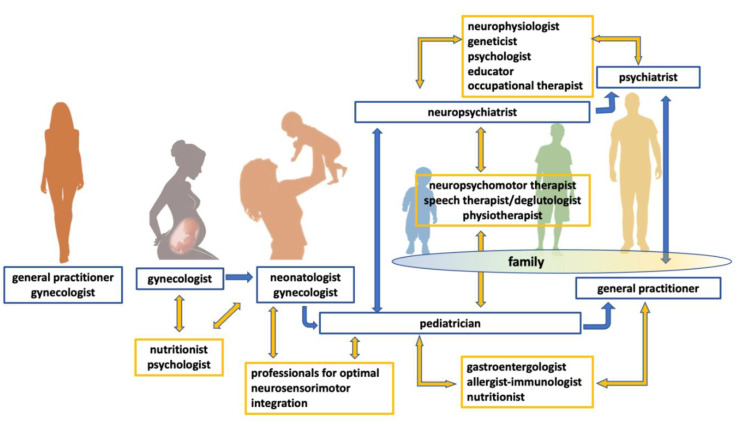
Mother’s health is the premise for a successful intrauterine life. During pregnancy, the gynecologist ensures the best control of risk factors and the enhancement of protective factors, largely related to proper maternal nutrition and supplementation. After birth, the baby–mother dyad is supported by the neonatologist–gynecologist duo. The qualified support for the well-being of the mother is integrated by neonatal care, which includes the best conditions of neurosensory-motor integration aimed at the physiological postnatal neuronal wiring. With variable times and modalities—mostly depending on the outcome of the pregnancy and the characteristics of the newborn—the child’s care is subsequently entrusted to the pediatrician, who provides suggestions for a positive physical and neuro-psychomotor development. In case of clinical abnormalities, the pediatrician prompts diagnostic pathways and early interdisciplinary interventions based on clinical and laboratory findings. In the event of motor and/or socio-communicative abnormalities, the pediatrician consults the neuropsychiatrist, who shall consider whether to include standardized diagnostic tools for ASD in the clinical assessment. The diagnosis of ASD is followed by further diagnostic evaluations (neurophysiologist, geneticist), functional assessment, and timely psychoeducational evidence-based interventions. The neuropsychiatrist orchestrates the cooperation of numerous professionals (psychologist, educator, occupational therapist), monitors the results, and tailors the supports according to the evolving skills and needs. In parallel with the neuropsychiatrist, the pediatrician prompts a clinical assessment according to the expected comorbidities in ASD, involving in particular gastroenterologist, allergist-immunologist, and nutritionist. A close collaboration with the neuropsychiatrist allows the best integration of physical and neuropsychiatric aspects, involving at the same time professionals linked both to the pediatric and the neuropsychiatric sides. The transition towards adulthood requires a handover on both levels of intervention, from the pediatrician to the general practitioner for the biological features, and from the neuropsychiatrist to the psychiatrist for the psychiatric sphere. The connection between the two levels (body and mind) is maintained even in adulthood. It should be noted that the above described structured model acquires worth and meaning if it places in the center the person with ASD and his/her family, as the main stakeholders of a flexible model, able to adapt to the evolving needs and favoring the highest level of feasible well-being.

## Data Availability

Not applicable.

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
