# Peer review of "Autism Spectrum Disorder from the Womb to Adulthood: Suggestions for a Paradigm Shift"

_jpm, 2021, doi:10.3390/jpm11020070_

Round 1

Reviewer 1 Report

I was wondering if the number of authors is not a problem. However, I think that numerous authors from excellent departments offer an interdisciplinary look at the problem of autism.

Similar to the authors, I accept the approach that the most effective intervention in ASD is expected by primary prevention aimed at pregnancy and at early control of the main effector molecular pathways. It is a correlation of all potential factors, such as microbiology, immunology, genetics, epigenetic factors, and living environment. I am a supporter of this approach and it seems to be the most appropriate in my opinion.

The quality of the manuscript is excellent. The authors provided crucial information in the article, including current SARS-coV-2 virus problem. I am delighted with the amount and reliability of the facts about autism presented. The text is clear and orderly.

The number of references is impressive and needed in such a comprehensive description of the ASD problem.

I only have a few minor comments:

- Please, correct the font in the authors' affiliations.

- Please, refine the description under Figure 1.

- Please, replace the spelling, e.g. "First 1000 Days" with First 1000 Days.

Reviewer 2 Report

This is a very interesting and well-written paper

I have a few comments:

Unfortunately, pages are not numbered so I will use pdf pages

Page 2, intro, typo “evidences”

Page 3, typo “neurodevelopmental disorders seems”

Page 3, 1.1, ASD is diagnosed by age 3 yo (at least onset of symptoms by); regression can happen outside of year #2

Page 4, typo: “ASD makes it implausible the hypothesis”

Add the literature of folate receptor autoantibodies during pregnancy to the 1.4 section would strengthen the paper – this is the key missing feature of this review and it potentially treatable

Page 9, microbiota discussion – it would strengthen the paper to discuss the increased risk of ASD after C-section (this is only briefly mentioned on page 11).

Page 10, section 1.7 – it would strengthen the paper to list areas of neuroinflammation that have been correlated with ASD symptoms (Rossignol and Frye, 2014  Front Physiol 5:150   Evidence linking oxidative stress, mitochondrial dysfunction, and inflammation in the brain of individuals with autism)

Page 11, leaky gut – better to list this as increased intestinal permeability

Page 11, butyrate—would be helpful to mention the literature on this being lower in ASD

Page 11, MTHFR risk – there are at least 3 meta-analysis which would be better references (Razi, et al., 2020  Research in Autism Spectrum Disorders 70:101473   Association between MTHFR gene polymorphism and susceptibility to autism spectrum disorders: Systematic review and meta-analysis)

Page 13, references 197 and 198 would add reference: Walker, et al., 2019  Sci Rep 9(1):5987   A molecular biomarker for prediction of clinical outcome in children with ASD, constipation, and intestinal inflammation

Page 16, typo: ““bad trio” - involving maternal dysbiosis, immune activation ad oxidative stress”
